# Cross-regularization:
# Adaptive Model Complexity through Validation Gradients

**Carlos Stein Brito** [1]

## Abstract

Model regularization requires extensive manual tuning to balance complexity against overfitting. Cross-regularization resolves this tradeoff by directly adapting regularization parameters through validation gradients during training. The method splits parameter optimization - training data guides feature learning while validation data shapes complexity controls - converging provably to cross-validation optima. When implemented through noise injection in neural networks, this approach reveals striking patterns: unexpectedly high noise tolerance and architecture-specific regularization that emerges organically during training. Beyond complexity control, the framework integrates seamlessly with data augmentation, uncertainty calibration and growing datasets while maintaining single-run efficiency through a simple gradient-based approach.

## 1. Introduction

Regularization in machine learning requires careful tuning of parameters that control model complexity. Cross-validation provides only discrete feedback through training multiple models. This applies across different types of regularization - from classical norm-based penalties to modern approaches like stochastic regularization and data augmentation. While methods like variational approaches (Molchanov et al., 2017) attempt to learn regularization during training, they optimize proxy objectives on training data rather than directly targeting generalization performance. Previous approaches either required inverse Hessians (Larsen et al., 1998) or parameter history tracking (Maclaurin et al., 2015), or multiple runs (Jaderberg et al., 2017).

We introduce cross-regularization, a direct optimization of

[1]NightCity Labs, Lisbon, Portugal. Correspondence to: Carlos Stein Brito <carlos.stein@nightcitylabs.ai>.

*Proceedings of the 42nd International Conference on Machine Learning*, Vancouver, Canada. PMLR 267, 2025. Copyright 2025 by the author(s).

complexity controls through gradient descent on validation data. The method splits both parameters and data: model parameters optimize training loss while regularization parameters receive unbiased gradient information from a separate validation set. This provides continuous feedback about generalization during training. We prove convergence to cross-validation solutions under standard optimization assumptions, making the method theoretically grounded.

The framework applies naturally across regularization types. For norm-based regularization, we demonstrate automatic discovery of optimal penalty strength without hyperparameter search. In neural networks, learnable noise scales reveal distinct regularization phases tied to feature learning and architecture-specific patterns. The method extends to data augmentation policy learning, uncertainty calibration, and online adaptation to growing datasets, demonstrating broad applicability while maintaining single-run efficiency.

## 2. Background and Related Work

### 2.1. Validation-Gradient Optimization

Validation-gradients emerged as a principled approach to hyperparameter optimization (Bengio, 2000), computing gradients of validation performance by backpropagating through the training procedure. While theoretically elegant, these methods require either computing inverse Hessians (Larsen et al., 1998) or maintaining the full parameter update history (Maclaurin et al., 2015), limiting their applicability to modern networks.

Luketina et al. (Luketina et al., 2016) addressed the scaling challenge by computing validation gradients for hyperparameters from only the most recent parameter update. However, training regularization parameters in both data partitions prevented any convergence guarantees. Our work reformulates the validation gradient approach by directly optimizing the quantities controlled by hyperparameters - weight norms, noise scales and other complexity controls. This eliminates hyperparameters while maintaining the framework of validation-guided optimization. Other works have also utilized validation gradients, for example, to weight source tasks in transfer learning (Chen et al., 2022).

## 2.2. Approaches to Regularization

Traditional regularization methods like weight decay (Krogh & Hertz, 1992) and dropout (Srivastava et al., 2014) use fixed parameters that must be tuned through cross-validation (Stone, 1974). For modern networks, complexity controls often need to vary across components (Zhang et al., 2021), and other factors like the behavior of normalization layers under distribution shifts (Nado et al., 2020) or the geometry of the loss landscape (Foret et al., 2021) also make manual tuning prohibitive. This has led to adaptive methods that attempt to learn regularization parameters during training, like Concrete Dropout (Gal et al., 2017) and variational dropout (Molchanov et al., 2017). However, these approaches optimize proxy objectives on training data rather than directly targeting generalization, and can struggle to learn strong regularization due to their training dynamics.

Population Based Training (PBT) (Jaderberg et al., 2017) addresses generalization by adapting hyperparameters using validation performance, but requires training multiple parallel models. For noise regularization, Noh et al. (Noh et al., 2017) developed gradient-based noise optimization but without validation gradients. While methods like variational dropout (Molchanov et al., 2017) offer another adaptive approach by learning noise scales through Bayesian variational inference, they often struggle in practice due to optimization challenges and restrictive prior assumptions.

## 2.3. Statistical Learning Theory

Our theoretical analysis builds on Vapnik's structural risk minimization principle (Vapnik, 1999), which formalizes the tradeoff between empirical risk and model complexity. Cross-regularization provides direct optimization of this tradeoff through validation gradients. For linear models, optimal regularization relates directly to noise-to-signal ratios (Hastie et al., 2009), while in neural networks, recent work connects regularization schemes to network capacity (Bartlett et al., 2017). The interaction between regularization and optimization shapes which solutions gradient descent discovers (Bishop, 1995; Saxe et al., 2014).

## 3. Cross-regularization Method

Consider the standard machine learning setup where we want to minimize test loss while preventing overfitting through regularization. The traditional cross-validation approach involves training multiple models with different regularization strengths $\lambda$:

$$\min_\lambda \mathcal{L}_{\text{val}}(w^*(\lambda)) \tag{1}$$

$$\text{where} \quad w^*(\lambda) = \arg\min_w \{\mathcal{L}_{\text{train}}(w) + \lambda R(\rho(w))\}$$

for a $\lambda$-scaled penalty that depends on the parameters $w$ through $\rho(w)$, which we denote as regularization parameters and a penalty function $R(\cdot)$. For instance, in L2 regularization, we have $\rho(w) = |w|_2$ and $R(x) = x^2$. With cross-regularization, we demonstrate that the relevant regularization parameters $\rho$ can be directly optimized through gradient descent, eliminating indirect hyparameters $\lambda$.

### 3.1. General Framework

Cross-regularization separates model training into two parallel optimizations:

- Feature learning (model parameters $\theta$) on training data

- Complexity control (regularization parameters $\rho$) on regularization data

Throughout this work, we use a single train-validation split, though the method naturally extends to k-fold schemes. We denote the complexity training data as the regularization set, instead of the validation set, to clarify that it is used during training.

Concretely, for a model $f_{\theta,\rho}(x)$ with parameters $\theta$ and regularization parameters $\rho$, the training loss for features and validation loss for complexity are:

$$\mathcal{L}_{\text{train}}(\theta, \rho) = \mathbb{E}_{(x,y)\sim\mathcal{D}_{\text{train}}}[\ell(f_{\theta,\rho}(x), y)] \tag{2}$$

$$\mathcal{L}_{\text{val}}(\theta, \rho) = \mathbb{E}_{(x,y)\sim\mathcal{D}_{\text{val}}}[\ell(f_{\theta,\rho}(x), y)] \tag{3}$$

These are optimized through alternating updates:

$$\theta_{t+1} = \theta_t - \eta_\theta \nabla_\theta \mathcal{L}_{\text{train}}(\theta_t, \rho_t) \tag{4}$$

$$\rho_{t+1} = \rho_t - \eta_\rho \nabla_\rho \mathcal{L}_{\text{val}}(\theta_{t+1}, \rho_t) \tag{5}$$

for learning rates $\eta_\theta$ and $\eta_\rho$. This algorithm directly optimizes regularization strength while maintaining the separation between feature learning and complexity control. The validation gradients give $\rho$ continuous feedback about generalization, unlike the discrete feedback in cross-validation.

### 3.2. L2 Regularization Analysis

To demonstrate cross-regularization, we analyze it in the context of ridge regression where the relationship between regularization and solution complexity is well understood. The standard approach controls this complexity indirectly through a regularization parameter $\lambda$:

$$w^*(\lambda) = \arg\min_w \{\|Xw - y\|^2 + \lambda\|w\|_2^2\} \tag{6}$$

For each $\lambda$, this yields an optimal solution $w^*(\lambda)$ with corresponding norm $\rho^*(\lambda) = \|w^*(\lambda)\|_2$. As $\lambda$ increases, $\rho^*(\lambda)$ decreases monotonically - higher regularization enforces

smaller norms. This means searching over $\lambda$ to minimize validation loss:

$$\min_\lambda \mathcal{L}_{\text{val}}(w^*(\lambda)) \qquad (7)$$

is equivalent to directly searching over achievable norms $\rho^*$:

$$\min_{\rho^*} \mathcal{L}_{\text{val}}(w^*) \quad \text{subject to} \quad \|w^*\|_2 = \rho^* \qquad (8)$$

Rather than searching over $\lambda$, we can directly optimize the L2 constraint by reparameterizing the weights into magnitude and direction:

$$\rho = \|w\|_2, \qquad (9)$$
$$\theta = w/\|w\|_2 \quad \text{where} \quad \|\theta\|_2 = 1 \qquad (10)$$

This naturally separates the optimization problem 1:

$$\mathcal{L}_{\text{train}}(\theta, \rho) = \min_{\theta, |\theta|=1} \|\rho X\theta - y\|_{\text{train}}^2 \qquad (11)$$
$$\mathcal{L}_{\text{val}}(\theta, \rho) = \min_\rho \|\rho X\theta - y\|_{\text{val}}^2 \qquad (12)$$

**Theorem 3.1** (L2 Cross-Validation Equivalence). *Under smoothness and strong convexity conditions, cross-regularization, in eqs. 4 and 5, converges to the same solution as optimal ridge regression:*

$$\rho^* \theta^* = w_{val}(\lambda^*)$$

*where $w_{val}(\lambda^*)$ is the ridge solution with best validation $\lambda$. The general proof for convex losses appears in appendix.*

Since optimizing validation loss over $\lambda$ is equivalent to optimizing over achievable norms $\rho^*$, we can directly optimize $\rho^*$ through gradient descent rather than searching over $\lambda$. This eliminates the indirect hyperparameter while maintaining explicit control over model complexity. The optimization naturally decomposes into two subproblems: an outer loop that optimizes $\rho^*$ using validation loss, and an inner loop that optimizes over the space of parameters orthogonal to the norm constraint. In other words, for each fixed $\rho^*$, the inner optimization learns the optimal direction in parameter space while maintaining the prescribed complexity.

Rather than solving these nested optimization problems exactly, cross-regularization performs alternating gradient updates using different losses for each parameter set. While similar in structure to coordinate descent, this approach differs fundamentally by using separate objective functions: training loss for direction parameters and validation loss for norm parameters. This split optimization allows continuous adaptation of model complexity during training, providing immediate feedback about generalization performance rather than waiting for complete convergence at each complexity level.

## 3.3. Parameter Partition through Gradient Decomposition

Cross-regularization separates model parameters from regularization parameters. However, for non-smooth penalties like L1 norms, no natural parameter split exists. This limitation suggests shifting from separating parameters to separating parameter update directions - identifying subspaces that control model capacity versus feature learning.

This geometrical view leads to decomposing parameter gradients into orthogonal components:

$$g = g_\rho + g_\perp, \quad g_\rho \perp g_\perp \qquad (13)$$

For a regularizer $R(w)$, the regularization component captures movement in the direction of steepest complexity change, $\nabla_w R(w)$:

$$g_\rho = \text{Proj}_{\nabla R}(g) = \frac{\nabla_w R}{|\nabla_w R|_2} \cdot (g^T \nabla_w R) \qquad (14)$$

For L1 regularization, $\nabla_w R = \text{sign}(w)$ yields:

$$g_{\text{L1}} = \tilde{u}_{\text{L1}} \cdot (g^T \tilde{u}_{\text{L1}}), \quad \tilde{u}_{\text{L1}} = \frac{\text{sign}(w)}{|\text{sign}(w)|_2} \qquad (15)$$

Training updates occur in the complementary subspace through $g_\perp$, maintaining current complexity, while validation updates through $g_\rho$ modulate regularization strength. This geometric perspective generalizes to any differentiable regularizer by identifying its characteristic complexity direction.

## 3.4. Stochastic Regularization

Modern neural networks rely heavily on stochastic regularization, where random perturbations during training improve generalization (Bishop, 1995). These perturbations can take many forms - from additive noise to random masking - but all require careful tuning of their magnitude, as it determines the strength of regularization: higher values force more robust features at the cost of computational precision. While manual tuning can find a single optimal noise level, cross-validation becomes impractical when different parts of the model require different noise scales.

A simple univariate example illustrates how validation gradients can automate noise tuning. Consider a Gaussian model for regression, $p(y|x) = \mathcal{N}(wx, \sigma^2)$, trained on a single data pair $(x_t, y_t)$. Training both $w$ and $\sigma$ on this data leads to overfitting ($\sigma \to 0$), but using a validation point $(x_v, y_v)$ to tune $\sigma$ while training $w$ recovers appropriate uncertainty.

This principle scales to neural networks through noise scale parameters optimized by validation gradients. However, noise typically reduces accuracy unless we treat the model as sampling from a distribution - similar to ensemble methods

or Bayesian inference. By averaging predictions during validation, the method can discover noise levels that improve generalization. Thus we define the inference at validation using Monte Carlo averaging:

$$f_{\text{val}}(x) = \mathbb{E}_\epsilon[f(x, \epsilon)] \approx \frac{1}{K} \sum_{k=1}^{K} f(x, \epsilon_k), \quad \epsilon_k \sim \mathcal{N}(0, \sigma)$$

(16)

Leading to distinct objectives for training and validation:

$$\mathcal{L}_{\text{train}} = \mathbb{E}_{(x,y) \sim \mathcal{D}_{\text{train}}} \mathbb{E}_\epsilon[\ell(f(x, \epsilon), y)] \tag{17}$$

$$\mathcal{L}_{\text{val}} = \mathbb{E}_{(x,y) \sim \mathcal{D}_{\text{val}}}[\ell(\mathbb{E}_\epsilon[f(x, \epsilon)], y)] \tag{18}$$

Training with single noise samples maintains the regularizing effect of randomization, while validation averages multiple samples to measure generalization. Note that a deterministic validation scheme ($\epsilon = 0$) would make the validation loss independent of noise scales $\sigma$, preventing their optimization.

# 4. Theoretical Analysis

Four theoretical results establish the core properties of cross-regularization: the alternating updates converge linearly, the optimization landscape admits local analysis, the statistical complexity scales with regularization parameters, and the method achieves cross-validation performance. All proofs appear in the Appendix.

Our first theoretical result establishes convergence of parameter updates despite the differing training and regularization objectives. Under standard smoothness and strong convexity conditions, the alternating optimization scheme, eqs. 4 and 5, achieves linear convergence:

**Theorem 4.1** (Linear Convergence). *For appropriate learning rates, cross-regularization, eqs. 4 and 5, converges linearly:*

$$\|\theta_t - \theta^*\|^2 + \|\rho_t - \rho^*\|^2 \leq (1-\kappa)^t (\|\theta_0 - \theta^*\|^2 + \|\rho_0 - \rho^*\|^2)$$

*where $\kappa > 0$ depends on problem constants.*

The linear convergence holds despite the distinct objectives for $\theta$ and $\rho$.

Neural networks and other complex models have nonconvex loss landscapes. To address this, we first characterize convergence through local geometry.

**Theorem 4.2** (Local Structure). *Let $L(\theta, \rho)$ be twice continuously differentiable in a neighborhood of a local minimum $(\theta^*, \rho^*)$ with Hessian $H$. If $L$ has $L_H$-Lipschitz continuous Hessian, then there exists a radius $r = \min\left(\frac{\mu}{6L_H}, \frac{(1-\gamma)\mu}{2\|H\|}\right)$ (where $\mu = \lambda_{\min}(H)$, $\gamma \in (0, 1)$ is a constant determining the preserved fraction of strong convexity, and $\|H\|$ is the spectral norm) such that in the ball $B_r(\theta^*, \rho^*)$, the loss*

*admits the decomposition $L(\theta, \rho) = L(\theta^*, \rho^*) + \frac{1}{2}(\mathbf{z} - \mathbf{z}^*)^T H(\mathbf{z} - \mathbf{z}^*) + R(\theta, \rho)$, where $\mathbf{z} = (\theta, \rho)^T$ and the remainder $R$ satisfies $\|R(\theta, \rho)\| \leq \frac{\gamma\mu}{2}(\|\theta - \theta^*\|^2 + \|\rho - \rho^*\|^2)$.*

The local quadratic structure implies a well-behaved loss landscape within this radius, providing the smoothness and effective strong convexity crucial for stable optimization. In non-convex settings like neural networks, stability hinges on managing interactions between model parameters ($\theta$) and regularization parameters ($\rho$). Our neural network analysis (Theorem 4.3) assumes conditions like Lipschitz continuous validation loss gradients with respect to model parameters. These assumptions ensure bounded coupling, preventing conflicting gradients from destabilizing $\rho$ optimization, a behavior empirically supported by stable noise adaptation over extended training (Appendix G).

Building on local geometric insights, we further establish convergence guarantees for neural networks under practical assumptions:

**Theorem 4.3** (Convergence for Neural Networks). *Under the assumptions that: (1) the training dynamics of model parameters $\theta$ converge to a stationary point $\theta^*(\rho)$ for any fixed regularization parameters $\rho$; (2) the validation loss $\mathcal{L}_{extval}(\theta, \rho)$ is $\alpha$-strongly convex in $\rho$ for any fixed $\theta$; and (3) the gradients $\nabla_\rho \mathcal{L}_{extval}(\theta, \rho)$ are $\beta$-Lipschitz continuous with respect to $\theta$, then the alternating optimization scheme of cross-regularization (Eqs. 4-5) converges to a stationary point $(\theta^*, \rho^*)$.*

Thus, Theorems 4.2 and 4.3 establish that cross-regularization mirrors the convergence behavior of fixed-regularization models, even within the non-convex landscapes of neural networks.

## 4.1. Statistical Rate

The dimensionality of regularization parameters $\rho$ ($k \ll d$) is typically much smaller than model parameters $\theta$ (dimension $d$). This low-dimensional structure leads to favorable statistical bounds:

**Theorem 4.4** (Statistical Rate). *The regularization parameters converge to the population optimum at rate:*

$$\|\rho_m - \rho_{true}^*\|^2 \leq O\left(\frac{k \log(1/\delta)}{m}\right)$$

*where $m$ is the regularization set size.*

The $O(\sqrt{k/m})$ convergence rate directly reflects the method's efficiency - statistical error scales with the few regularization parameters ($k$) rather than the full model dimension ($d$).

Cross-regularization also matches the performance of standard cross-validation:

**Theorem 4.5** (Cross-validation Equivalence). *Under mild conditions on the regularizer, cross-regularization achieves the same validation loss as the optimal cross-validation solution.*

Both approaches minimize validation loss over identical solution spaces but through different parameterizations. The theory thus confirms cross-regularization's practical advantages: linear convergence in optimization, statistical scaling with only regularization parameters, and recovery of cross-validation optima.

## 5. Norm-Based Regularization Examples

### 5.1. L2 Regularization

To validate the theoretical analysis, we construct synthetic data where regularization is essential for generalization. From a small set of independent base features, we generate a larger feature set by adding correlated variants with Gaussian noise. This design creates groups of highly correlated features, making the linear system ill-conditioned - without regularization, the model can exploit these correlations to overfit by assigning large opposing weights to related features.

The empirical results in Figure 1(A-C) show cross-regularization converging to optimally tuned ridge regression through direct gradient descent. The optimization reveals an inherent tension in regularization - training loss increases as the method reduces complexity to improve validation performance, demonstrating how validation gradients guide the tradeoff between memorization and generalization, reducing the generalization gap (difference between train and test accuracy).

### 5.2. L1 Regularization

L1 regularization presents a fundamental challenge: the non-differentiability of the L1 norm prevents direct norm decomposition. This is particularly important for medical applications where feature selection helps identify relevant biomarkers. On the diabetes progression prediction task, standard LASSO requires evaluating an entire regularization path to find the optimal sparsity level.

Training updates follow $g_\perp$ while validation updates control sparsity through $g_{L1}$. This allows simultaneous optimization of prediction accuracy and feature selection without manual tuning. Figure 1(D) shows cross-regularization automatically discovers optimal sparsity levels matching LASSO's best cross-validated performance.

### 5.3. Derivative Norm for Splines

Smoothing splines measure complexity through integrated squared derivatives, which for B-spline bases reduces to a quadratic form $|D\beta|^2$ with finite difference matrix $D$. The gradient decomposition handles this matrix norm by projecting updates onto level sets of constant smoothness:

$$g_\rho = \text{Proj}_{D^T D\beta}(g) = \frac{D^T D\beta}{|D^T D\beta|_2} \cdot (g^T D^T D\beta) \quad (19)$$

Training follows $g_\perp$ to maintain smoothness while validation gradients through $g_\rho$ adapt complexity to local feature scales, as demonstrated in Figure 1(E). This shows how the gradient decomposition framework naturally accommodates differential regularizers.

## 6. Neural Network Regularization

### 6.1. Interpretable Noise Regularization

Cross-regularization allows noise parameters at any granularity, from per-unit to global. We demonstrate the approach using layer-wise noise as it balances adaptivity to network structure against statistical efficiency - while per-unit noise offers maximal flexibility, it introduces $O(d^2 L)$ regularization parameters, and global noise cannot capture architectural differences. Layer-wise noise requires only $O(L)$ parameters while respecting natural network boundaries.

Cross-regularization requires only differentiable noise parameters, making it compatible with multiplicative Gaussian noise but not standard Dropout, though both share similar regularization properties (Srivastava et al., 2014; Molchanov et al., 2017). While multiplicative noise offers scale invariance, we implement additive noise after normalization to allow for direct analysis of noise structure:

$$u_l = w_l^T h_{l-1} \quad (20)$$
$$\hat{u}_l = (u_l - \mu_u)/\sigma_u \quad (21)$$
$$h_l = g(\hat{u}_l + \sigma_l \epsilon), \quad \epsilon \sim \mathcal{N}(0, I) \quad (22)$$

for weights $w_l$, nonlinearity $g(\cdot)$, and Layer normalization, chosen over Batch normalization to avoid confounding regularizing effects of batch stochasticity. This design ensures noise magnitudes remain interpretable across layers, though multiplicative noise achieves similar results with less interpretable scales (see Appendix). During training, we use single noise samples, switching to averaged samples for validation and deterministic prediction $f(x, \epsilon = 0)$ at test time. While L2 regularization is a common technique, we focused on noise-based regularization for our neural network experiments given its prominence and effectiveness in regularizing deep neural networks.

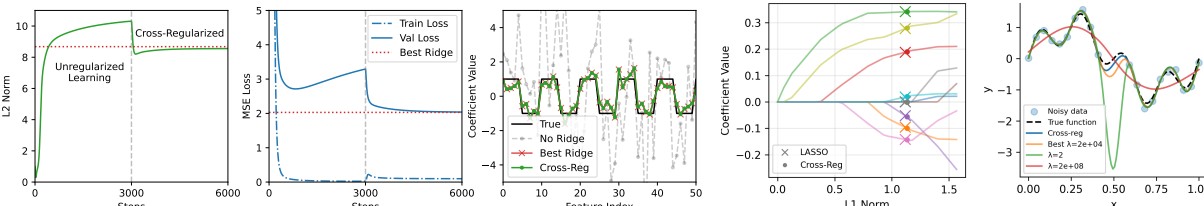

Figure 1. Cross-regularization results across different norms. (A-C) L2 regularization: Evolution of weight norm showing adaptation to optimal ridge levels, training/validation loss dynamics, and recovered coefficient values matching cross-validated ridge regression. (D) L1 regularization on diabetes prediction: Validation loss versus L1 norm shows automatic discovery of optimal sparsity matching LASSO. (E) Spline smoothing: Learned function achieves appropriate complexity without manual tuning.

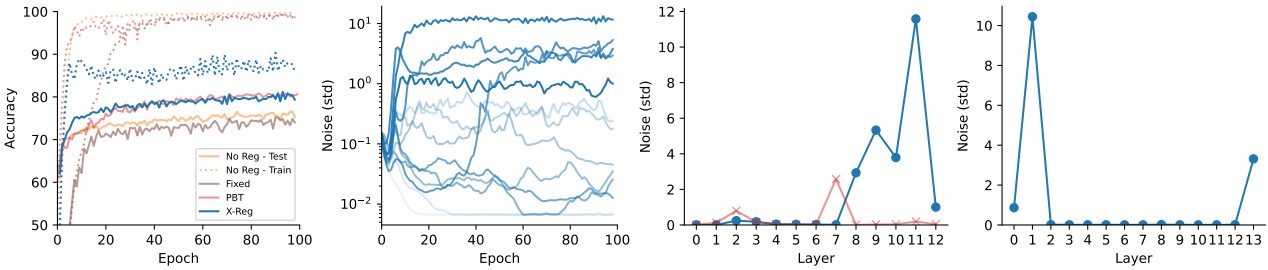

Figure 2. Noise dynamics in VGG-16 on CIFAR-10 reveal architectural regularization patterns. (A) Cross-regularization matches PBT accuracy (83.7%) while improving over baseline (76.0%) in a single training run. Fixed noise ($\sigma = 1$ in the final five layers) impedes initial learning yet fails to prevent later overfitting. (B) Layer noise adaptation tracks overfitting - low during feature learning, then increasing to prevent memorization, when a generalization gap appears (around epoch 5), and again when the training accuracy rises again (around epoch 40), after a period of strong regularization. (C) Final noise distribution (0.01-12 $\sigma$) shows unprecedented yet functional noise levels, surpassing PBT's more conservative regime (max 2.8 $\sigma$). (D) ResNet noise concentrates in early and final layers, with 14.9 $\sigma$ in layer 2. Such a profile enforces a bottleneck at points where there is high information flow.

---

**Algorithm 1** Cross-regularization Training

1: **Input**: Parameters $\theta$, $\rho$; datasets $\mathcal{D}_{\text{train}}$, $\mathcal{D}_{\text{reg}}$
2: **for** each epoch **do**
3:     **for** $(x, y) \sim \mathcal{D}_{\text{train}}$ **do**
4:         $\epsilon \sim \mathcal{N}(0, I)$
5:         $\mathcal{L}_{\text{train}}(f_{\theta,\rho}(x, \epsilon), y)$
6:         Update $\theta$ using gradient descent
7:         **if** every reg_interval steps **then**
8:             $(x_r, y_r) \sim \mathcal{D}_{\text{reg}}$
9:             $\hat{y} = \frac{1}{K} \sum_{k=1}^{K} f_{\theta,\rho}(x_r, \epsilon_k)$
10:           $\mathcal{L}_{\text{reg}}(\hat{y}, y_r)$
11:           Update $\rho$ using gradient descent
12:         **end if**
13:     **end for**
14: **end for**

---

### 6.2. Adaptive Noise Dynamics

The layer-normalized design provides standardized signal-to-noise ratios, allowing direct analysis of information capacity (Fig. 2). Noise patterns expose computational structure: early layers preserve precise feature detection while deeper layers force increasingly robust representations. When validation accuracy plateaus, indicating potential overfitting, noise selectively increases in layers prone to memorization.

Cross-regularization leads to a regime where neural networks remain functional under surprisingly high noise levels - up to 13 standard deviations post-normalization. While these levels are far beyond conventional regularization strengths, their emergence through gradient-based optimization of generalization performance offers insights into network robustness and capacity.

Our noise level $\sigma$ can be related to standard dropout rates $p$ through signal-to-noise ratio analysis. For layer-normalized activations with unit variance, additive Gaussian noise $\mathcal{N}(0, \sigma^2)$ gives SNR $= 1/\sigma^2$. Standard dropout with rate $p$, after the $1/(1 - p)$ scaling, yields unit variance noise at $p = 0.5$ (equivalent to $\sigma = 1$). For comparison, typical dropout rates like $p = 0.2$ correspond to $\sigma \approx 0.5$. In

contrast, our observed noise level of $\sigma = 12$ corresponds to dropout rates of $p = 0.987$ - revealing a previously unexplored high-noise regime that only becomes accessible through gradual adaptation, as directly initializing with such extreme noise levels would prevent learning.

This high-noise regime, while initially appearing to dramatically challenge assumptions about neural robustness, finds an interesting parallel in the neural network pruning literature, particularly the Lottery Ticket Hypothesis (Frankle & Carbin, 2019). Our observed noise levels of $\sigma \approx 13$ create a significant information bottleneck. Quantitatively, this level of noise corresponds to an information capacity of approximately $0.004$ bits per symbol, which is mathematically equivalent to pruning with a sparsity of around $99.4\%$. This aligns remarkably well with findings from the Lottery Ticket Hypothesis, which has shown that VGG-style networks on CIFAR-10 can be pruned to similar sparsity levels (e.g., $98\%$) while maintaining performance, often with sparsity concentrated in later layers, similar to our noise patterns. This convergence of findings from two distinct methodologies—gradient-based noise adaptation versus iterative weight pruning—suggests that cross-regularization is effectively identifying intrinsic properties of neural information capacity and architecture-specific compressibility.

This adaptive mechanism offers a valuable window into generalization dynamics. Rather than prescribing fixed regularization schedules, noise levels automatically track each layer's capacity to overfit. The gradual emergence of extreme noise demonstrates networks can learn robust features even under severe perturbation when guided by validation gradients. Furthermore, these adaptively learned noise levels and the resulting model performance remain stable over extended training periods, as confirmed by simulations run for 600 epochs (see Figure 13 in Appendix G).

Comparison with Population-Based Training (PBT) validates these findings - both methods optimize noise using validation performance, but through different mechanisms. While evolutionary search also discovers layer-wise patterns but more conservative magnitudes, gradient-based optimization reveals higher functional noise levels, and with an order of magnitude less computation, requiring only $O(T(1 + K/r))$ forward passes versus PBT's $O(PT)$.

Fixed noise injection ($\sigma = 1$ in final five layers) illustrates the limitations of static regularization: too strong initially, slowing feature learning, yet insufficient to prevent later overfitting, exhibiting larger generalization gaps. We also attempted a comparison with variational dropout (Molchanov et al., 2017) but found it unsuitable for layer-wise noise adaptation. Despite extensive hyperparameter tuning of the KL weight, the method either collapsed to zero noise or became unstable - precisely the kind of manual tuning our approach aims to eliminate.

## 6.3. Additional Architectures

To validate our method's generality across architectures, we apply cross-regularization to a ResNet with noise injection (WideResNet-16-4). The noise adaptation reveals a striking pattern: high noise emerges in both early and final layers, with $10.4\ \sigma$ the first layer ($\sigma_1 = 0.9, \sigma_2 = 10.4, \sigma_{14} = 3.3$), while maintaining low noise in middle layers (Fig. 2-D).

This pattern reflects the network's architecture - since skip connections allow information to bypass middle layers, the network concentrates regularization at early layers that process all inputs and final layers that integrate features, creating an information bottleneck that can't be bypassed. These results demonstrate how cross-regularization discovers complexity controls that reflect network topology. The concentration of noise in layers that cannot be circumvented by skip connections provides evidence for how residual architectures shape information flow and regularization requirements. This finding connects to theoretical work showing residual networks can be viewed as ensembles of paths of different lengths (Veit et al., 2016).

## 6.4. Parameter Sensitivity Analysis

Empirical studies validate cross-regularizations robustness across hyperparameters (Appendix F). While single samples suffice during training, validation requires 3-5 MCMC samples for stable gradient estimation, with performance deteriorating below this range (Fig. 11). Regularization updates can be sparse (every 30 steps) with minimal impact on convergence, resulting in only $10\%$ computational overhead. The method maintains performance with extremely small regularization sets - down to $1\%$ of training data (Fig. 12), aligning with our statistical analysis that error scales with regularization dimension $k$ rather than model dimension $d$ (Theorem 4.3). These results establish an efficient configuration for practical use.

## 6.5. Automatic Uncertainty Calibration

Reliable uncertainty estimation is critical for deploying machine learning systems in practice. Medical diagnosis requires accurate confidence scores to determine when to defer to human experts, autonomous systems need calibrated uncertainties for safe decision making, and active learning systems rely on uncertainty estimates to select informative samples. Current approaches either require post-hoc corrections (Guo et al., 2017), model-specific assumptions (Gal et al., 2017), or separate training objectives (Lakshminarayanan et al., 2017). However, real-world applications need models that provide reliable uncertainties immediately upon deployment and adapt these estimates as they continue learning.

Analysis of the neural network experiments reveals that

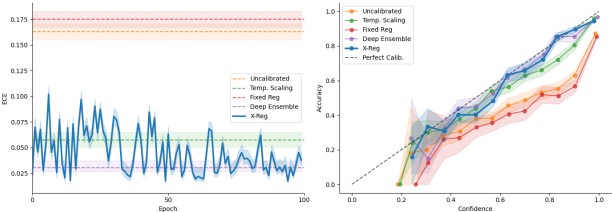

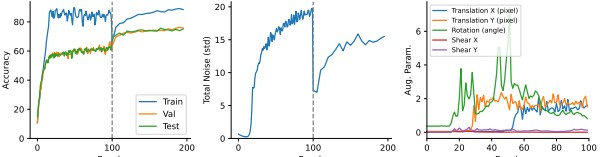

*Figure 3.* Uncertainty calibration and ECE evolution. Left: Expected Calibration Error (ECE) over training epochs for Cross-Regularization (X-Reg), compared to final ECE values for an uncalibrated model, Temperature Scaling, a Fixed Regularization model (Fixed Reg), and a Deep Ensemble. Right: Reliability diagram comparing these methods. Shaded areas represent 95% confidence intervals. X-Reg maintains strong calibration throughout training and performs competitively against strong baselines.

*Figure 4.* Dataset growth adaptation and adaptive augmentation. A: Performance evolution shows successful knowledge transfer at epoch 100 transition from partial to full dataset. B: Total regularization strength automatically adapts - stronger regularization compensates for limited initial data, then decreases as full dataset provides natural regularization. Vertical line marks dataset transition. C: Evolution of learned augmentation parameters on SVHN. Translation parameters (pixels) and rotation angles (degrees) increase early in training before stabilizing, while shear transformations remain minimal. Results demonstrate automatic discovery of dataset-specific invariances favouring rigid transformations.

cross-regularization achieves strong calibration. Figure 3 illustrates this by comparing Cross-Regularization (X-Reg) against several baselines: an uncalibrated model, Temperature Scaling (Guo et al., 2017), a model with manually fixed regularization parameters (Fixed Reg), and a 5-member Deep Ensemble. The right panel shows the reliability diagrams, where X-Reg closely tracks perfect calibration. The left panel demonstrates the Expected Calibration Error (ECE) evolution for X-Reg alongside the final ECE values for the baselines. X-Reg achieves a final ECE of 0.038 (with 79.5% accuracy), showcasing strong calibration in a single training run that is competitive with the 5-member Deep Ensemble (ECE 0.030, Acc. 81.3%), and significantly surpassing the uncalibrated model (ECE 0.163, Acc. 67.4%), Temperature Scaling (ECE 0.057, Acc. 69.6%), and the Fixed Reg model (ECE 0.175, Acc. 74.7%). Shaded areas in the figure represent 95% confidence intervals. This online calibration represents a fundamental advance, as the model maintains calibrated uncertainties even as it learns, without requiring a separate post-hoc calibration phase.

This automatic online calibration represents a fundamental advance in uncertainty estimation. The adaptive noise scales learned through validation simultaneously control model complexity and shape predictive uncertainty, enabling immediate deployment with reliable confidence scores. As the model encounters new data, both its predictions and uncertainty estimates adapt naturally without requiring recalibration or retraining. This direct connection between validation performance and uncertainty quantification suggests cross-regularization learns to modulate predictions based on their empirical reliability, providing a practical solution for systems requiring trustworthy real-time uncertainty estimates.

### 6.6. Adaptive Regularization Under Data Growth

Training data often becomes available incrementally, requiring models to learn from limited data while adapting as samples accumulate. While increasing dataset size improves generalization (Nakkiran et al., 2020), optimal regularization typically requires manual adjustment with data growth. We study adaptation to growing datasets by training initially on 20% of data before incorporating the full dataset. This mirrors practical scenarios where models must deploy with limited data while preparing for growth.

Results demonstrate automatic adaptation to dataset size (Figure 4-A,B). During limited-data training, the model maintains generalization through elevated regularization. Upon transitioning to the full dataset, regularization adjusts downward while preserving performance, aligning with theoretical understanding that larger datasets require less explicit regularization. The smooth adaptation suggests applications to continual learning settings where data distributions evolve over time.

### 6.7. Adaptive Data Augmentation

Data augmentation through label-preserving transformations forms a cornerstone for regularization in modern deep learning, yet tuning transformation magnitudes remains a manual, dataset-specific task. Methods like AutoAugment (Cubuk et al., 2019) and Population Based Augmentation (Ho et al., 2019) automate this search through reinforcement learning or evolution, but require thousands of training runs. While cross-regularization optimizes model parameters through validation gradients, applying this approach to data transformations appears problematic - random perturbations of inputs should degrade validation performance, pushing gradient descent to minimize all transformations.

Data augmentation fits naturally into our framework by

viewing transformations as a distribution over model configurations, analogous to noise-based regularization. Each transformed input represents a sample from the space of valid variations of the original image. For each type of transformation (e.g., rotation, translation), we define continuous magnitude parameters (e.g., for rotation, a learnable maximum angle $\alpha_{rot}$ defines a range $U[-\alpha_{rot}, \alpha_{rot}]$ from which a specific angle is sampled for each training instance). These magnitude parameters $\alpha$ become part of the model's optimizable regularization parameters $\rho$. As with noise regularization, we maintain the asymmetry between training and validation: single samples during training provide stochastic regularization, while Monte Carlo averaging during validation measures expected performance across transformations. This implementation follows the same alternating optimization algorithm as noise-based regularization: the transformation magnitude parameters $\alpha$ are updated using validation gradients $\nabla_\alpha \mathcal{L}_{\text{val}}$ (obtained through standard backpropagation as these parameters affect the validation loss), aiming to maximize generalization.

This validation-guided optimization creates a natural equilibrium for the transformation strengths. If transformations are too aggressive (e.g., rotating a digit '6' by 180 degrees to resemble a '9'), they will degrade validation performance, and the gradients will drive down their magnitudes. Conversely, if transformations are too mild to provide sufficient regularization, the model may overfit to the training data, leading to a higher validation loss compared to a more optimally regularized state, again guiding the parameters $\alpha$ towards more effective levels. The approach naturally encourages transformation-invariant features by penalizing models that fail to generalize across valid variations of the input. This formulation connects preprocessing parameters to stochastic parameters in Bayesian neural networks (Blundell et al., 2015; Wan et al., 2013), as both represent distributions over model configurations optimized through Monte Carlo sampling.

Figure 4-C demonstrates this adaptation on SVHN: translations converge to 1-2 pixels, rotations to 3 degrees, while shear transformations diminish. These learned parameters match the intuition that digit recognition benefits from rigid transformations over deformations, emerging automatically from validation gradients. The optimization improves test accuracy from 82.8% to 86.3%, while reducing the generalization gap from 16.2% to 7.3%. The smooth parameter trajectories throughout training reveal when different transformations become more or less useful for regularization. This optimization of preprocessing transforms through validation gradients points to applications in automating feature extraction and data preprocessing, while quantifying the relationship between datasets and their underlying invariances.

## 7. Conclusion

Cross-regularization demonstrates that model complexity control can be directly optimized through validation gradients rather than requiring manual tuning through cross-validation. This optimization enables continuous adaptation of regularization parameters during training, replacing discrete hyperparameter search. The method provides a unified optimization framework for different forms of regularization, from classical norm penalties to stochastic regularization and data augmentation.

Analysis through direct optimization reveals regularization to be a dynamic process that evolves with model training. In deep networks, regularization requirements reflect architectural structure, with patterns of noise tolerance emerging across layers. These findings connect network architecture to optimal regularization strategy, advancing our understanding of how network design influences learning and generalization.

The effectiveness of validation-gradient optimization for complexity control establishes regularization as a learnable component of model training. This formulation eliminates manual tuning while offering a principled approach to regularization design that adapts to specific architectures and tasks.

## Acknoledgements

This work was developed with the support of NCL agents.

## Impact Statement

This work introduces cross-regularization, a method to advance Machine Learning by enabling more automated and efficient control of model complexity for improved generalization. By directly optimizing regularization parameters, our approach can lead to more robust and reliable AI systems, reduce manual tuning efforts, and potentially lower computational costs. It also offers insights into model adaptation and can enhance trustworthiness through better uncertainty calibration.

The societal implications of cross-regularization are generally aligned with those of broader progress in AI. We believe this method contributes positively by promoting more principled and efficient model development. While the development of more capable AI always warrants careful consideration, this specific work does not introduce new ethical risks beyond those inherent in advancing AI capabilities.

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

# A. Theoretical Analysis

We analyze the convergence of our alternating optimization scheme where model parameters $\theta$ train on training data while regularization parameters $\rho$ optimize on a separate regularization set. We show that the split optimization converges despite using different objectives for each parameter set.

## A.1. Optimization Analysis

We first establish convergence of the alternating gradient scheme under standard smoothness and strong convexity conditions.

**Theorem A.1** (Linear Convergence). *Assume the loss $L(\theta, \rho)$ satisfies:*

1. *$\beta$-smoothness in both arguments:*

$$\|\nabla L(\theta, \rho) - \nabla L(\theta', \rho')\| \leq \beta(\|\theta - \theta'\| + \|\rho - \rho'\|)$$

2. *$\mu$-strong convexity in $\theta$ for any fixed $\rho$*

3. *$\alpha$-strong convexity in $\rho$ for any fixed $\theta$*

*Then the alternating updates:*

$$\theta_{t+1} = \theta_t - \eta_\theta \nabla_\theta L_{train}(\theta_t, \rho_t) \tag{23}$$

$$\rho_{t+1} = \rho_t - \eta_\rho \nabla_\rho L_{reg}(\theta_{t+1}, \rho_t) \tag{24}$$

*with learning rates:*

$$\eta_\theta \leq 1/\beta \tag{25}$$

$$\eta_\rho \leq \min(1/\beta, \mu\eta_\theta/(4\beta^2)) \tag{26}$$

*converge linearly:*

$$\|\theta_t - \theta^*\|^2 + \|\rho_t - \rho^*\|^2 \leq (1 - \kappa)^t(\|\theta_0 - \theta^*\|^2 + \|\rho_0 - \rho^*\|^2)$$

*where $\kappa = \min(\mu\eta_\theta/2, \alpha\eta_\rho)$.*

*Proof.* The proof analyzes each update step separately and then combines them to show overall convergence. Let $\theta^*(\rho)$ denote the minimizer of $L(\cdot, \rho)$ for fixed $\rho$.

**Step 1 ($\theta$-update):** By $\beta$-smoothness and optimality of $\theta^*(\rho_t)$:

$$L_{\text{train}}(\theta_{t+1}, \rho_t) \leq L_{\text{train}}(\theta_t, \rho_t) - (\eta_\theta - \eta_\theta^2\beta/2)\|\nabla_\theta L_{\text{train}}\|^2$$

By $\mu$-strong convexity in $\theta$:

$$\|\nabla_\theta L_{\text{train}}\|^2 \geq 2\mu(L_{\text{train}}(\theta_t, \rho_t) - L_{\text{train}}(\theta^*(\rho_t), \rho_t))$$

Combining with $\eta_\theta \leq 1/\beta$ gives:

$$\|\theta_{t+1} - \theta^*(\rho_t)\|^2 \leq (1 - \mu\eta_\theta)\|\theta_t - \theta^*(\rho_t)\|^2 \tag{1}$$

**Step 2 ($\rho$-update):** By similar arguments and smoothness of $\nabla_\rho L_{\text{reg}}$ with respect to $\theta$:

$$\|\rho_{t+1} - \rho^*\|^2 \leq (1 - \alpha\eta_\rho)\|\rho_t - \rho^*\|^2 + \beta^2\eta_\rho\|\theta_{t+1} - \theta^*(\rho_t)\|^2 \tag{2}$$

**Step 3 (Combined Progress):** Let $V_t = \|\theta_t - \theta^*(\rho_t)\|^2 + \|\rho_t - \rho^*\|^2$. Combining (1) and (2):

$$V_{t+1} \leq [(1 - \mu\eta_\theta) + \beta^2\eta_\rho]\|\theta_t - \theta^*(\rho_t)\|^2 + (1 - \alpha\eta_\rho)\|\rho_t - \rho^*\|^2$$

The condition $\eta_\rho \leq \mu\eta_\theta/(4\beta^2)$ ensures the $\theta$ progress term dominates the coupling cost:

$$(1 - \mu\eta_\theta) + \beta^2\eta_\rho \leq 1 - \mu\eta_\theta/2$$

Therefore:

$$V_{t+1} \leq (1 - \kappa)V_t$$

where $\kappa = \min(\mu\eta_\theta/2, \alpha\eta_\rho) > 0$

Iterating gives the final bound:

$$V_t \leq (1 - \kappa)^t V_0$$

$$\square$$

Note that the condition on learning rates arises from our chosen update order, suggesting this asymmetry is not fundamental to the method but rather an artefact of the ordering.

This result establishes that despite optimizing different objectives for $\theta$ and $\rho$, the alternating scheme converges linearly to the optimum.

### A.2. Local Approximation Analysis

Let $L(\theta, \rho)$ be twice continuously differentiable in a neighborhood of a local minimum $(\theta^*, \rho^*)$ with Hessian $H$ satisfying Assumption 1 (from main text, referring to positive definiteness and bounded coupling). Since $(\theta^*, \rho^*)$ is a local minimum, we have $\nabla L(\theta^*, \rho^*) = 0$. By Taylor's theorem with remainder:

$$L(\theta, \rho) = L(\theta^*, \rho^*) + \frac{1}{2}\begin{pmatrix} \theta - \theta^* \\ \rho - \rho^* \end{pmatrix}^T H \begin{pmatrix} \theta - \theta^* \\ \rho - \rho^* \end{pmatrix} + R(\theta, \rho) \tag{27}$$

For the function with $L_H$-Lipschitz continuous Hessian, the remainder term is bounded by:

$$\|R(\theta, \rho)\| \leq \frac{L_H}{6}\|(\theta - \theta^*, \rho - \rho^*)\|^3 \tag{28}$$

By the assumptions for Theorem 4.2 (main text), $H$ is positive definite with $\lambda_{\min}(H) \geq \mu > 0$, giving us:

$$\begin{pmatrix} \theta - \theta^* \\ \rho - \rho^* \end{pmatrix}^T H \begin{pmatrix} \theta - \theta^* \\ \rho - \rho^* \end{pmatrix} \geq \mu\|(\theta - \theta^*, \rho - \rho^*)\|^2 \tag{29}$$

For the quadratic approximation to maintain $\gamma\mu$-strong convexity (for some $\gamma \in (0, 1)$), as stated in Theorem 4.2, we require:

$$\|R(\theta, \rho)\| \leq \frac{(1 - \gamma)\mu}{2}\|(\theta - \theta^*, \rho - \rho^*)\|^2 \tag{30}$$

Combining with our bound on $R$:

$$\frac{L_H}{6}\|(\theta - \theta^*, \rho - \rho^*)\|^3 \leq \frac{(1 - \gamma)\mu}{2}\|(\theta - \theta^*, \rho - \rho^*)\|^2 \tag{31}$$

This yields $\|(\theta - \theta^*, \rho - \rho^*)\| \leq \frac{3(1-\gamma)\mu}{L_H}$. The radius $r$ in Theorem 4.2 is established by taking the minimum of this and another constraint related to Hessian approximation validity, $r = \min\left(\frac{\mu}{6L_H}, \frac{(1-\gamma)\mu}{2\|H\|}\right)$. Within this radius, the remainder term satisfies the condition in Theorem 4.2, ensuring that the function is effectively $\gamma\mu$-strongly convex, preserving local optimization properties.

## A.3. Statistical Analysis

Having established optimization convergence, we analyze how sample size affects the solution quality. The main result is that the statistical error scales with regularization parameter dimension $k$ rather than model dimension $d$.

**Theorem A.2** (Statistical Rate). *Assume:*

1. *Loss bounded:* $|L(z; \theta, \rho)| \leq B$

2. *Population risk $R$ is $\alpha$-strongly convex in $\rho$*

3. *Parameter spaces compact:* $\Theta \subset \mathbb{R}^d, \mathcal{P} \subset \mathbb{R}^k$

*Then with probability at least $1 - \delta$:*

$$\|\rho_m - \rho_{true}^*\|^2 \leq \frac{4B^2}{\alpha^2} \left( \frac{k \log(1/\delta)}{m} \right)$$

*where $\rho_{true}^*$ minimizes the population risk and $m$ is the regularization set size.*

*Proof.* By standard uniform convergence over the $k$-dimensional space $\mathcal{P}$:

$$\sup_{\rho \in \mathcal{P}} |R_m(\theta^*(\rho), \rho) - R(\theta^*(\rho), \rho)| \leq B\sqrt{\frac{k \log(1/\delta)}{m}}$$

Strong convexity then gives:

$$\|\rho_m - \rho_{\text{true}}^*\|^2 \leq \frac{2}{\alpha}(R(\rho_m) - R(\rho_{\text{true}}^*)) \leq \frac{4B^2}{\alpha^2} \left( \frac{k \log(1/\delta)}{m} \right)$$

$\square$

This $O(\sqrt{k/m})$ rate explains why small regularization sets suffice - the error depends on regularization parameter dimension $k$ (typically number of layers) rather than model dimension $d$ (total parameters).

## A.4. Cross-validation Equivalence

We show our direct optimization achieves the same solution as standard cross-validation. The proof relies on showing that both methods optimize over equivalent solution spaces through different parameterizations.

**Theorem A.3** (Cross-validation Equivalence). *Let $f(\rho)$ be a complexity measure. For any $\lambda > 0$, let $\theta_\lambda$ be the solution from cross-validation:*

$$\theta_\lambda = \arg\min_\theta \{\mathcal{L}_{train}(\theta) + \lambda f(\rho)\}$$

*Assume $\lambda$ is monotonic in the solution $\rho$. Then cross-regularization achieves the same validation loss as cross-validation:*

$$\min_\lambda \mathcal{L}_{val}(\theta_\lambda) = \min_\rho \mathcal{L}_{val}(\theta(\rho))$$

*Proof.* By monotonicity of $\lambda$ in $\rho$, we can rewrite cross-validation optimization:

$$\min_\lambda \mathcal{L}_{\text{val}}(\theta_\lambda) = \min_\rho \mathcal{L}_{\text{val}}(\theta_\lambda)$$

where $\theta_\lambda = \arg\min_\theta \{\mathcal{L}_{\text{train}}(\theta) + \lambda f(\rho)\}$

This is equivalent to direct optimization of $\rho$ in cross-regularization:

$$\min_\rho \mathcal{L}_{\text{val}}(\theta(\rho))$$

Therefore both methods optimize the same objective over the same solution space, just parameterized differently through $\lambda$ or direct optimization of $\rho$. $\square$

## A.5. Proof of Neural Network Convergence Stability

The theorem regarding the convergence of cross-regularization for neural networks (Theorem 4.3) is established under the following assumptions:

1. The training dynamics of $\theta$ converge to a stationary point $\theta^*(\rho)$ for any fixed $\rho$.

2. The validation loss $\mathcal{L}_{\text{val}}(\theta, \rho)$ is $\alpha$-strongly convex in $\rho$ for any fixed $\theta$.

3. The gradients $\nabla_\rho \mathcal{L}_{\text{val}}(\theta, \rho)$ are $\beta$-Lipschitz continuous with respect to $\theta$.

The proof, summarized below, demonstrates that for sufficiently small learning rates $\eta_\rho$, the error terms for both model and regularization parameters converge to zero:

*Proof.* Define the error metrics:

$$E_\theta(t) = \|\theta_t - \theta^*(\rho_t)\|^2 \tag{32}$$
$$E_\rho(t) = \|\rho_t - \rho^*\|^2 \tag{33}$$

By assumption 1, $\theta$ converges to $\theta^*(\rho)$ for any fixed $\rho$ with some convergence rate. When $\rho$ is updated, this introduces a perturbation in the optimization landscape. We can model this as:

$$E_\theta(t+1) \leq \gamma \cdot E_\theta(t) + \delta \cdot \|\rho_{t+1} - \rho_t\|^2 \tag{34}$$

where $\gamma < 1$ captures the convergence rate of $\theta$ and $\delta$ is a coupling constant that measures how changes in $\rho$ affect the optimization of $\theta$.

For the $\rho$ updates, we use strong convexity (assumption 2) and Lipschitz continuity (assumption 3):

$$\|\rho_{t+1} - \rho^*\|^2 \leq \|\rho_t - \eta_\rho \nabla_\rho \mathcal{L}_{\text{val}}(\theta_{t+1}, \rho_t) - \rho^*\|^2 \tag{35}$$
$$= \|\rho_t - \eta_\rho \nabla_\rho \mathcal{L}_{\text{val}}(\theta^*(\rho_t), \rho_t) - \rho^*\|^2 + \|\eta_\rho(\nabla_\rho \mathcal{L}_{\text{val}}(\theta_{t+1}, \rho_t) - \nabla_\rho \mathcal{L}_{\text{val}}(\theta^*(\rho_t), \rho_t))\|^2 \tag{36}$$
$$\leq (1 - \eta_\rho \alpha) E_\rho(t) + \eta_\rho^2 \beta^2 E_\theta(t+1) \tag{37}$$

Where we used the fact that for $\alpha$-strongly convex functions with step size $\eta_\rho \leq \frac{2}{\alpha}$, the update contracts the distance to the optimum by a factor of $(1 - \eta_\rho \alpha)$.

Combining these inequalities, we get a linear system:

$$\begin{bmatrix} E_\theta(t+1) \\ E_\rho(t+1) \end{bmatrix} \leq \begin{bmatrix} \gamma & \delta \eta_\rho^2 \\ \eta_\rho^2 \beta^2 & 1 - \eta_\rho \alpha \end{bmatrix} \begin{bmatrix} E_\theta(t) \\ E_\rho(t) \end{bmatrix} \tag{38}$$

Let's denote this matrix as $M$. For the system to converge, we need all eigenvalues of $M$ to have magnitude less than 1. The eigenvalues are the solutions to:

$$\det(M - \lambda I) = 0 \tag{39}$$

For sufficiently small $\eta_\rho$, we can ensure that both eigenvalues have magnitude less than 1, as the diagonal terms $\gamma$ and $(1 - \eta_\rho \alpha)$ are dominant. Specifically, when:

$$\eta_\rho < \min\left(\frac{2}{\alpha}, \sqrt{\frac{(1-\gamma)(1 - \eta_\rho \alpha)}{\delta \beta^2}}\right) \tag{40}$$

Then both error terms converge to zero as $t \to \infty$, establishing convergence to a stationary point $(\theta^*, \rho^*)$.  $\square$

# B. Norm based examples

## B.1. L2 Regularization Details

We validate L2 cross-regularization on synthetic data specifically designed to demonstrate the importance of regularization. From 5 independent base features, we create 100 total features by adding Gaussian noise ($\sigma = 0.1$) to copies of the base features. True coefficients alternate between +1 and -1 for features derived from the same base feature. This design creates groups of highly correlated features, making the linear system ill-conditioned. Without regularization, the model can exploit these correlations to fit noise by assigning large positive and negative weights to redundant features.

The cross-regularization model uses stochastic gradient descent with learning rates 0.01 for both feature learning and L2 regularization. Training runs for 6000 steps with regularization starting at step 3000 to demonstrate adaptation. For baseline comparison, we fit ridge regression models across 1000 logarithmically-spaced values of $\lambda$ from $10^{-3}$ to $10^{1}$.

## B.2. L1 Regularization Details

We evaluate L1 cross-regularization on the diabetes regression dataset (Efron et al., 2004). The dataset consists of 442 patients with 10 physiological features. Data is standardized and split 80/20 into train/validation sets.

The cross-regularization model uses stochastic gradient descent with momentum 0.99, learning rate 0.0005 for feature learning and 0.01 for L1 regularization. Training runs for 2000 epochs with batch size 512. For baseline comparison, we fit LASSO models across 50 logarithmically-spaced values of $\lambda$ from $10^{-2.5}$ to 1. Extended results are shown in Fig. 5.

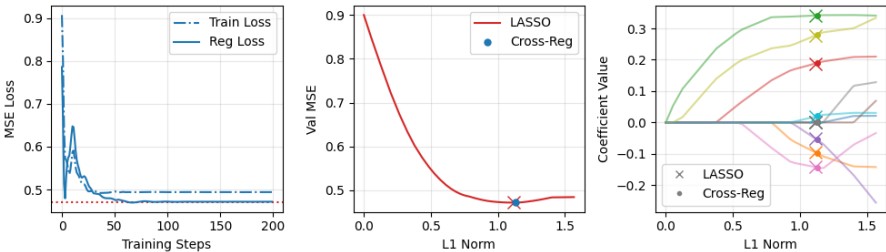

*Figure 5.* Extended L1 cross-regularization results on diabetes dataset. (A) Training dynamics demonstrating stable convergence despite non-smooth L1 penalty. (B) Validation MSE versus L1 norm comparing cross-regularization and LASSO. (C) Coefficient paths showing similar sparsity patterns were discovered through direct optimization rather than cross-validation.

## B.3. Spline Regularization Details

We generate synthetic data from the function:

$$y(x) = \sin(2\pi x) + 0.5\sin(8\pi x) + \epsilon(x)$$

with heteroskedastic noise $\epsilon(x) \sim \mathcal{N}(0, (0.5 + x)^2\sigma^2)$. This provides a challenging test case with both smooth and sharp features. We sample 20 points with gaps to evaluate interpolation.

The B-spline model uses cubic basis functions with 15 knots. Second derivatives are computed analytically and normalized by trace. Training uses SGD with learning rate 0.3 for both parameters and smoothness. Extended results shown in Fig. 6.

*Figure 6.* Extended Spline Results: (A) Validation loss evolution demonstrates convergence to cross-validation performance. (B) Smoothness norm adaptation reveals automatic discovery of appropriate complexity. (C) Comparison of fitted functions across different smoothness levels.

## C. Neural Network Implementation

Cross-regularization requires only two modifications to standard neural network training: adding noise parameters after normalization layers and implementing validation updates. The method can be implemented in a few lines of code, with no custom optimizers or complex architectural changes needed.

### C.1. Model Architecture

Each layer block in our neural networks applies normalization (LayerNorm or BatchNorm) followed by learnable noise:

- Linear / Convolutional layer
- Normalization (without affine)
- Additive noise with learned scale $\sigma_l = \exp(\rho_l)$
- ReLU activation

### C.2. Regularization Class

The algorithm requires only the separation of parameters and datasets. We implement this through an abstract `RegularizedModel` class, which must specify the set of regularization parameters $\rho$. The remaining parameters are considered training parameters. This abstraction facilitates models to implement different forms of regularization while maintaining the same training procedure.

### C.3. Training Protocol

Optimization settings:

- Adam optimizer
- Learning rates: $10^{-4}$ (model), $10^{-1}$ (noise)
- Initialization: $\log \sigma = -3$
- Batch size: 512
- Training epochs: 100

### C.4. Computational Analysis

Computational requirements for T training steps:

- Standard train-validation split: $O(T(1 + \frac{v}{1-v}))$ forward passes, for a v validation split %, (1-v) test split %, and one validation run per training epoch. For v=10%: $O(1.11T)$.
- Cross-regularization: $O(T(1 + \frac{K}{r}))$ forward passes for $K$ MCMC samples and regularization update interval $r$. For K=3 and r=30: $O(1.1T)$.

- M-fold cross-validation: $O(MT)$ forward passes.
- PBT: $O(PT)$ forward passes. Memory usage: O(PM) for model size M and population size P.

## D. Population-Based Training

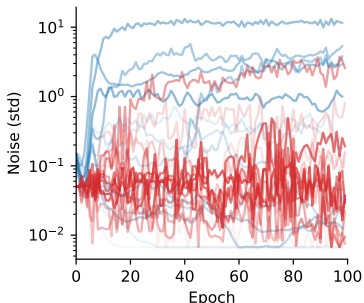

*Figure 7.* Noise dynamics for PBT. Noise evolution in PBT exhibits more volatile adaptation compared to the smooth emergence in cross-regularization, reflecting the discrete nature of evolutionary updates.

PBT was implemented with a population size of 20 models, using validation accuracy for selection and random perturbation factors in [0.8, 1.2] for noise parameter updates. The population size was chosen to balance computational cost with optimization stability.

The evolutionary optimization in PBT leads to distinct training dynamics compared to cross-regularization's gradient-based approach. Figure 7 shows the more volatile noise adaptation, with sudden changes in noise levels corresponding to selection and mutation events.

## E. ResNet Experiments

WideResNet-16-4 implements residual blocks with dual 3x3 convolutions and batch normalization across three stages. The architecture's skip connections create an ensemble of paths with varying depths (Veit et al., 2016), enabling multiple information routes. Our experiments reveal two findings: first, despite injecting noise levels comparable to VGG-16, the network maintains performance through its skip connections; second, the training dynamics (Figure 8) show distinct noise adaptation phases - an initial increase at epoch 5 when validation plateaus, followed by a second rise at epoch 40 as training accuracy approaches 100%. This pattern mirrors our CIFAR-10 results, reinforcing the connection between generalization gaps and noise adaptation.

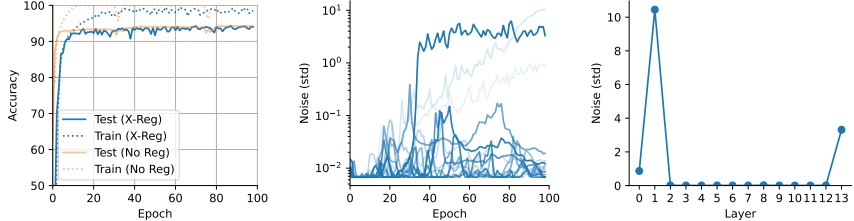

*Figure 8.* ResNet training dynamics: (a) Generalization gap emerges at epoch 5 and training nearly overfits at epoch 40, triggering corresponding noise adaptations. (b) Layer-wise noise evolution follows VGG-16 pattern, increasing at validation plateaus and overfitting. (c) Final noise concentrates in layers 1, 2 and 14, exceeding 10 standard deviations without compromising performance.

### E.1. BatchNorm and Multiplicative Noise

The standard ResNet implementation with BatchNorm and multiplicative noise exhibits similar qualitative dynamics, though with less interpretable scales (reaching over 3000 standard deviations)

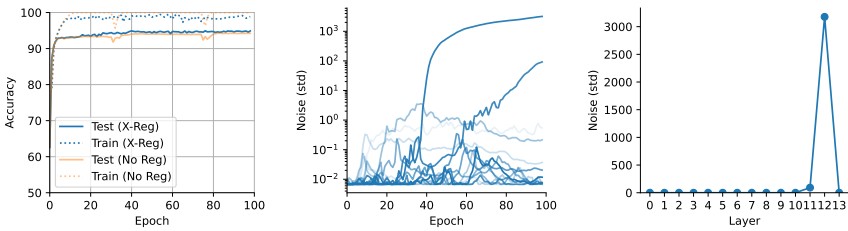

*Figure 9.* Training dynamics for ResNet with BatchNorm and multiplicative noise.

# F. Parameter Sensitivity Analysis

Figures 10, 11 and 12 show the detailed plots for the systematic method analyses.

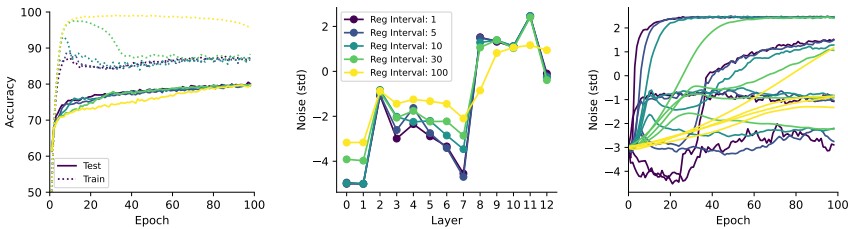

*Figure 10.* Effect of update interval. Consistent noise patterns emerge despite different update frequencies. Smoother evolution with more frequent updates but comparable convergence.

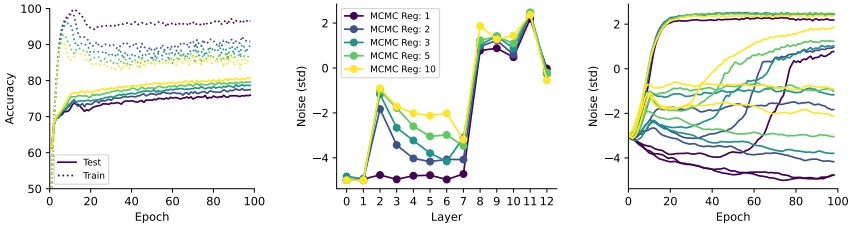

*Figure 11.* Impact of MCMC samples for regularization. Accuracy with different sample counts shows diminishing returns beyond 3-5 samples.

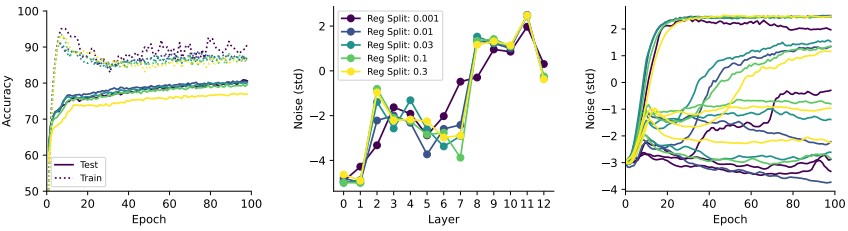

*Figure 12.* Sensitivity to regularization set size. Performance remains stable down 1% of training data. This efficiency stems from the low-dimensional nature of the regularization parameters compared to model weights.

# G. Extended Stability Simulations

To further demonstrate the stability of the learned regularization parameters in neural networks, Figure 13 shows the evolution of layer-wise noise sigmas ($\sigma_l$) over an extended training period of 600 epochs. The noise parameters stabilize

after an initial adaptation phase, confirming the empirical robustness of the adaptive mechanism discussed in Subsection 6.2.

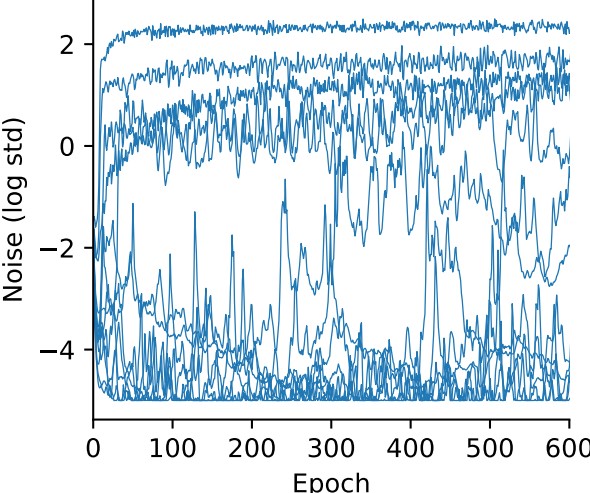

*Figure 13.* Evolution of layer-wise noise sigmas ($\sigma_l$) over 600 training epochs for a VGG-style network on CIFAR-10. The regularization parameters demonstrate long-term stability after an initial adaptation period.

