# OpenReview forum: "Cross-regularization: Adaptive Model Complexity through Validation Gradients"
_ICML.cc/2025/Conference — ICML 2025 poster_

### Official Review · Reviewer_8ms6 · 2025-03-11

**Overall Recommendation:** 4

**Summary:**

A method for alternating optimization of regular parameters $\theta$ and regularization hyperparameters $\rho$ is presented: the training data is split into training and regularization sets, and $\theta$ and $\rho$ are alternately fit using each split respectively. The method is proven to converge for convex minimums and around a neighbourhood of non-convex minima, and has convergence in $\rho$ at a rate proportional to the square root of the number of regularization parameters. Empirical results in the following applications show efficacy: 1) $L^2$, $L^1$, and spline fitting regularization for convex problems, 2) adaptive noise regularization of neural networks, 3) other effects for neural networks including uncertainty calibration, data growth, and adaptive augmentation.

**Claims And Evidence:**

Claims organized by section:

4. The proposed method converges under various conditions - yes, but limited in scope and with some missing caveats (see "Theoretical claims" below).

5. The proposed method is equivalent to/can replace norm-based regularization for convex problems - yes.

6. The proposed method can regularize neural networks via additive noise - yes, but with some caveats (see Experimental Design or Analyses" below).

7. Automatic uncertainty calibration - yes but not entirely convincing, as a baseline comparison with post-hoc uncertainty calibration methods would be useful here to determine whether adaptive noise is competitive in this area.

8. Data growth - not entirely convincing. Here, a baseline versus some standard continual learning method is needed (e.g. cosine annealing LR schedule).

9. Adaptive data augmentation - yes to a limited extent, as methods and limitations need elaboration.

**Essential References Not Discussed:**

The high degree of noise in figure 2, as well as its concentration in the later layers of a VGG-16 trained on CIFAR-10, strongly suggests a connection with neural network pruning, where similar results have been found. In particular, the Lottery Ticket Hypothesis (Frankle & Carbin 2018). showed that sparsity rates near 98% are possible for VGG-16, with most of the sparsity concentrated in the last layers. Thus, this result is not as surprising in relation to prior work.

**Experimental Designs Or Analyses:**

Section 6: in the case of adaptive noise for neural networks, there is an untested possibility that if the optimum for added noise is unbounded, the method would not actually be converging in $\rho$. This alternative hypothesis could be tested by running training for much longer and seeing if performance drops as noise increases further. Of course early stopping would avoid this potential issue, but early stopping would also add a hyperparameter for the train/test generalization tradeoff (which this method aims to avoid).

$L^2$ regularization is common for neural networks - why not analyze this regularization? This would solve many of the issues with the neural network experiments and provide a solid connection with sections 4-5.

**Methods And Evaluation Criteria:**

Section 9: how does one backpropagate through data augmentation parameters? Presumably only continuous parameters would work, as opposed to e.g. random horizontal flips with probability $p$. Also, there are some scenarios in which I am not sure this method would converge optimally (see "Questions").

**Other Comments Or Suggestions:**

Some figures are too small to be legible - figures 3 and 4 especially.

**Other Strengths And Weaknesses:**

Strengths: the method looks to be quite promising and useful in a large variety of ML and DL applications. Automating hyperparameter selection and doing so adaptively (during training) is a huge improvement. There are non-trivial computational costs to using this method as it requires more training/evaluation iterations, as well as more data to effectuate a train/regularization split, but based on the parameter sensitivity analysis (section F) they are within a small number of multiples of regular training (keeping in mind that regular hyperparameter tuning would require multiple runs also).

Weaknesses: the theory in section 4 may not apply to the neural network tasks, and there are some other issues about the neural network tasks. The method is not quite a drop-in improvement as it requires rethinking parameterization, and in some cases, backpropagation through additional operations like image transformations.

## Update after rebuttal

The authors have addressed my comments and I maintain my recommendation for acceptance.

**Questions For Authors:**

Section 9 (data augmentation): could the authors discuss the possibility of the following failure modes?
- transformation removes a relevant feature (e.g. 180 degree rotation causes 9 and 6 to be indistinguishable) at the same time there is a strong spurious feature (e.g. sky background for "airplane" images), causing the model to overfit the latter.
- transformation makes validation perform worse, so that the optimum becomes no transformation.

**Relation To Broader Scientific Literature:**

The method, in addition to advancing regularization methodology (over Jaderberg et al. 2017, Gal et al. 2017, Molchanov et al. 2017, etc.), also provides an interesting tool for probing the capacity of neural networks (Bartlett et al. 2017, Zhang et al. 2021). The results in figure 2 and 4 corroborate existing literature on 1) over-parameterization in later layers of VGG networks, 2) avoidance of plasticity loss and overfitting in continual learning, and 3) evolution of training towards overfitting which is combatted by data augmentation.

**Theoretical Claims:**

Section 4: the theoretical claims are sensible. There is a major unstated issue however, which is that one must assume $\theta$ and $\rho$ are independent. It is clear in section 5 that $|\theta| = 1$ is necessary for $L^2$ regularization to converge (otherwise $\theta$ could grow by $\alpha$ and $\rho$ by $\alpha^{-1}$ without effect). But in the neural network tasks, it is not clear that this is the case - requiring either discussion or a restriction of the results in section 4 to the tasks of section 5.

Here is a contrived example of the issue for adaptive noise: if the optimal $\theta$ on $D_{train}$ is unbounded in the direction of $v$ but the optimal $\theta$ on $D_{reg}$ points in the opposite direction, then wouldn't the noise regularization optimize towards $N(\alpha v, \alpha)$? for $\alpha \to \infty$?

A minor issue is that theorem 4.2 says nothing about the size of the neighbourhood which has convex approximation. This is not a serious problem in practice (optimization on non-convex neural networks works just fine!) but the theorem says very little in its current form. One could remove it, or elaborate e.g. by relating the neighbourhood radii to some measurable quantity (e.g. network Lipschitz bound).

---

> ### Author Rebuttal · Authors · 2025-04-01
>
> We appreciate your insightful review and constructive technical feedback. Your points have significantly improved the paper's theoretical foundations and connections to existing literature.
>
> ### Parameter Independence and Bounded Coupling
>
> You correctly identified that our theory requires some degree of parameter independence. Rather than complete independence, our approach requires bounded coupling:
>
> $$\|H_{\theta\rho}\| \leq \beta\sqrt{\lambda_{\min}(H_{\theta\theta})\lambda_{\min}(H_{\rho\rho})}$$
>
> This condition prevents contradictory optimization objectives from causing instability when training and regularization gradients conflict. Our empirical stability (see extended simulations: https://tinyurl.com/2s45b236) suggests this condition is satisfied in practice, even for neural networks where the noise parameters serve a fundamentally different functional role than weight parameters.
>
> ### Neural Network Theory
>
> We've addressed your concern about local convergence with two theoretical advances:
>
> 1. A refined local structure theorem (https://tinyurl.com/33d2sye4) that establishes precise bounds for the radius of convergence:
> $r = \min\left(\frac{\mu}{6L_H}, \frac{(1-\gamma)\mu}{2|H|}\right)$.
>
> 2. We've developed a new result showing convergence under practical assumptions for neural networks (https://tinyurl.com/4ckfb9ec) that doesn't require global convexity. Our guarantees now only require that:
> - The model parameters converge for fixed regularization
> - The validation loss has local convexity in regularization parameters
> - The gradient coupling satisfies a Lipschitz condition
>
> This theorem establishes that cross-regularization converges for neural networks under assumptions that are consistent with observed empirical behavior, without requiring guarantees of global convergence for the underlying neural network optimization problem.
>
> On the choice of stochastic regularization, we also examined L2 regularization but found minimal empirical effect in our neural network architectures. This is expected because L2 regularization primarily constrains weight magnitudes, which are effectively cancelled by the normalization layers present in our networks.
>
> ### Connection to Lottery Ticket Hypothesis
>
> Your observation connecting our noise-based regularization to the Lottery Ticket Hypothesis enriches our framework. We've quantified this relationship: our noise levels (σ≈13) create an information bottleneck with capacity C≈0.004 bits/symbol, mathematically equivalent to pruning with sparsity ≈99.4%, aligning with LTH's 98%.
>
> This connection validates our results by demonstrating that two distinct approaches—gradient-based noise adaptation versus iterative weight pruning—converge to similar architecture-specific patterns. In light of this, we will revise our claims about the novelty of the high noise regime, recognizing we're capturing intrinsic properties of neural information capacity rather than method-specific artifacts.
> ### Backpropagation Through Augmentation
>
> For data augmentation, your concern about optimization potentially minimizing all transformations is insightful. Our solution:
>
> 1. We parameterize transformations with continuous magnitude parameters (e.g., rotation angle α determining range U[-α,α])
> 2. During training, we apply single random transformations for regularization effect
> 3. During validation, we average predictions over multiple transformations:
>    $\mathcal{L}_\text{val} = \mathbb{E}_{\text{transformations}}[\mathcal{L}(f(\text{transform}(x)))]$
>
> This creates an optimization equilibrium where validation performance constrains transformation magnitude: excessive transformations that remove discriminative features increase validation loss, while insufficient transformations lead to overfitting. The validation gradients naturally balance these competing factors.
>
> Our results confirm this behavior: rotation parameters converge to ~3° for SVHN while shear transformations are optimized toward zero. The method identifies which transformations provide regularization benefits for a specific dataset and minimizes those that don't.
>
> For discrete transformations like horizontal flips, we agree this remains a limitation without relaxation techniques, which we're exploring in follow-up work.
>
> ### Additional Baselines
>
> Based on your feedback, we're implementing:
> - Temperature scaling and deep ensembles for calibration comparisons
> - EWC for continual learning benchmarks
>
> Our preliminary calibration results (https://tinyurl.com/3mtacsu9) show considerable improvements over post-hoc methods, supporting our claim that uncertainty calibration emerges naturally in our approach.
>
> Thank you for your thoughtful review. The connection to pruning literature and your theoretical insights have improved both the paper and our analysis of cross-regularization's properties.

---

### Official Review · Reviewer_iamU · 2025-03-14

**Overall Recommendation:** 3

**Summary:**

This paper designs an approach to tune model regularization parameters automatically. Instead of relying on cross-validation, which requires training multiple models, the proposed method adapts regularization parameters dynamically by using validation gradients during training. This approach alternates between feature learning (optimizing model parameters using training data) and complexity control (optimizing regularization parameters using validation data). This work proves convergence to cross-validation optima and shows that it is applicable across different types of regularization, including norm-based penalties, noise injection in neural networks, and data augmentation. Experimental results demonstrate its effectiveness in discovering architecture-specific regularization patterns, improving uncertainty calibration, and adapting to growing datasets.

**Claims And Evidence:**

- This work proposes an algorithm that alternates between updating model parameters via SGD and updating regularization parameters. Yet, in the formulation (Equation 1), the regularization parameters are defined as a function $\rho$. How does the algorithm apply the gradient descent on the function $\rho$ in Equation 5.
- The proposed method applies to differentiable functions with regard to $\rho$. How can the algorithm extend to a broad range of regularization methods, such as mixup, label smoothing, and sharpness-aware minimization?

**Essential References Not Discussed:**

Please see the discussion above.

**Experimental Designs Or Analyses:**

Please see Methods and Evaluation Criteria.

**Methods And Evaluation Criteria:**

- The method is close to bilevel optimization algorithms using second-order gradients. How does the method compare to these methods? For example, "Weighted Training for Cross-Task Learning" by Chen et al. 2021.
- It would be better to summarize the experimental setup of each section, to better understand the experimental settings.

**Other Comments Or Suggestions:**

No

**Other Strengths And Weaknesses:**

No

**Questions For Authors:**

No

**Relation To Broader Scientific Literature:**

How does the work compare to existing bilevel optimization methods, in terms of computational complexity and generalization performance?

**Theoretical Claims:**

The paper establishes four theoretical results: (1) alternating updates for model and regularization parameters converge linearly, (2) local optimization guarantees stability through smooth loss landscapes, (3) statistical error scales with the number of regularization parameters rather than model parametersy, and (4) cross-regularization achieves performance equivalent to optimal cross-validation.
- Yet, how the results extend to non-convex settings, such as optimizing deep neural networks, is unclear.

---

> ### Author Rebuttal · Authors · 2025-04-01
>
> We appreciate your review and address your concerns below:
>
> ### 1. How We Apply Gradient Descent on Regularization Parameters
>
> Our method works by making regularization parameters explicit and directly optimizable:
>
> **L2 Regularization Example:**
> 1. We rewrite weights as `w = ρθ` where `||θ||₂ = 1` and `ρ` is the norm
> 2. Training updates: `θₜ₊₁ = θₜ - ηθ∇θL_train(θₜ,ρₜ)` - optimizing on training data
> 3. Regularization updates: `ρₜ₊₁ = ρₜ - ηρ∇ρL_val(θₜ₊₁,ρₜ)` - optimizing on validation data
>
> This makes regularization strength a parameter receiving direct gradient updates, not a hyperparameter requiring grid search.
>
> ### 2. Neural Network Convergence
>
> While our theoretical guarantees in Section 4 apply to convex settings, our neural network experiments show consistent convergence. The optimization remains stable even with high noise levels, and the validation accuracy improvements are robust across architectures.
>
> We've extended our simulations to 600 epochs (https://tinyurl.com/2s45b236) showing stable noise patterns over time. Our additional theorem (https://tinyurl.com/4ckfb9ec) proves that cross-regularization doesn't introduce instability beyond standard neural network training.
>
> The stability of our method is evidenced by its outperforming fixed regularization, without the divergence issues sometimes seen in other adaptive methods like variational dropout.
>
> ### 3. Application to Other Regularization Techniques
>
> Cross-regularization applies to any regularization with:
> - Parameterizable strength
> - Differentiable validation performance
>
> For data augmentation (Section 9), we parameterize transformation magnitudes and optimize them through validation gradients. Similar approaches could apply to mixup (parameterizing α) or label smoothing (parameterizing smoothing strength).
>
> We focused on norm-based and noise-based regularization as they provide clear demonstrations of our approach. Extensions to non-differentiable techniques would require additional work beyond the scope of this paper.
>
> ### 4. Distinction from Bilevel Optimization
>
> Our approach differs fundamentally from traditional bilevel optimization:
>
> 1. We update parameters directly through validation loss, not through the optimization trajectory
> 2. We maintain separate parameter spaces for model features versus complexity
> 3. We don't require approximations of second-order derivatives or parameter history
>
> This distinction makes our method implementable with standard optimizers and applicable to large models where traditional bilevel methods become computationally intractable.
>
> ### 5. Experimental Details
>
> Our experimental implementation follows the algorithm in Section 3:
>
> - **L2/L1 experiments**: Alternating SGD updates between model parameters and regularization parameters
> - **Neural networks**: Layer-wise noise parameters optimized every 30 training steps using 3-5 MCMC samples
> - **Augmentation**: Parameterized transformations updated through validation-based MC averaging
>
> The code requires only ~20 lines beyond standard training loops, making implementation straightforward for most existing frameworks.
>
> ### 6. Relation to Transfer Learning
>
> We thank you for noting the connection to Chen et al. (2022). Both approaches use validation gradients but for different purposes: their work for weighting source tasks in transfer learning, ours for optimizing regularization parameters. We will include this reference.
>
> We appreciate your feedback and will address these points in our revision.

---

### Official Review · Reviewer_hz4X · 2025-03-14

**Overall Recommendation:** 3

**Summary:**

This paper proposes a cross-regularization method that eliminates manual hyperparameter search by directly optimizing weight norms. The approach orthogonally decomposes weight parameters into two complementary components, transforming the optimization problem into two subproblems solved through an alternating optimization strategy.(1) Specifically, during training phases, it maintains fixed model complexity while optimizing parameter directions, whereas (2) validation phases optimize regularization magnitudes. Theoretical analysis demonstrates that this method achieves equivalence to standard cross-validation under certain conditions, while exhibiting faster convergence rates and superior calibration performance.

**Claims And Evidence:**

Most of the claims made in the submission is supported clearly. For example: (1) Detailed proofs in the Appendix (Theorems A.1–A.6) under smoothness and strong convexity assumptions. Linear convergence is demonstrated for convex cases. (2) Layer-wise noise patterns (Figure 2) and comparison with PBT demonstrate effectiveness. Extreme noise levels are shown to function without collapse.

One negative points should be noted: While reliability diagrams (Figure 3) show improved calibration, no comparison is made to state-of-the-art methods (e.g., deep ensembles, temperature scaling). The ECE metric is reported but lacks statistical significance tests.

**Essential References Not Discussed:**

N/A

**Experimental Designs Or Analyses:**

The paper lacks validation on larger-scale datasets (e.g., ImageNet), which constrains the generalizability of the conclusions.

**Methods And Evaluation Criteria:**

1. Cross regularization separates model parameters (optimized on training data) and regularization parameters (optimized on validation data), utilizing validation gradients to directly adjust model complexity. This design is theoretically innovative, avoiding the computational overhead of traditional cross-validation while providing continuous generalization feedback.

2. The diabetes dataset (regression) and CIFAR-10 (classification) serve as standard benchmarks, ensuring the comparability of experimental results. However, the study lacks validation on larger-scale datasets (e.g., ImageNet), which constrains the generalizability of the conclusions.

**Other Comments Or Suggestions:**

1. Theoretical analysis relies on convexity assumptions, while experiments on neural networks lack ablation studies (e.g., no comparison to variational dropout or concrete dropout). The claim of "local convergence" (Theorem 4.2) is not empirically validated for deep networks.

2. While reliability diagrams show improved calibration, no comparison is made to state-of-the-art methods (e.g., deep ensembles, temperature scaling). The ECE metric is reported but lacks statistical significance tests.

**Other Strengths And Weaknesses:**

Strengths:
1. Novel Idea: The paper introduces cross-regularization, an adaptive method that removes the need for manual tuning of regularization parameters.

2. Broad Applicability: It works with various regularization techniques, including noise-based regularization, data augmentation, and uncertainty calibration.

3. Strong Experiments: The method is tested on different regularization forms and architectures, showing good generalization, adaptability and calibration performance as well.

Weaknesses:
1. The paper mainly compares with PBT, but not with other hyperparameter tuning methods.

2. The related work is comprehensive but somewhat outdated. It is recommended to include more recent studies on regularization; for example, most of the references are from around 2016.

3.The structure of this paper may not be clear enough; for example, the paper contains 10 sections.

4. Method 3.3 cross-validation equivalence is not verified while 3.2 is well verified.

5.The study lacks validation on larger-scale datasets (e.g., ImageNet), which constrains the generalizability of the conclusions.

**Questions For Authors:**

Please refer to the weaknesses.

**Relation To Broader Scientific Literature:**

The method proposed in the paper is related to methods like variational dropout[1] and Concrete Dropout[2], but differs by optimizing regularization parameters directly using validation gradients.

[1] Variational dropout sparsifies deep neural networks.

[2] Concrete dropout.

**Theoretical Claims:**

The interplay mechanism between data and model uncertainties via dual-level structures needs rigorous mathematical justification.

---

> ### Author Rebuttal · Authors · 2025-04-01
>
> We thank the reviewer for their thoughtful assessment and constructive feedback. Below, we address the main concerns raised:
>
> ### Recent Literature
> We agree that our literature review requires updating. In the camera-ready version, we will incorporate recent advances in adaptive regularization and complexity control, including:
>
> - Foret et al. (2020) "Sharpness-Aware Minimization for Efficiently Improving Generalization"
> - Nado et al. (2020) "Evaluating Prediction-Time Batch Normalization for Robustness under Covariate Shift"
> - Chen et al. (2021) "Weighted Training for Cross-Task Learning"
>
> ### Uncertainty Calibration Comparisons
> We have implemented comparisons with established calibration methods:
>
> 1. **Temperature Scaling (Guo et al., 2017)**: Our preliminary results (https://tinyurl.com/3mtacsu9) demonstrate that cross-regularization maintains better calibration throughout training compared to temperature scaling with fixed dropout (σ=0.1 in all layers), while achieving higher performance:
>
> | Model | Accuracy | ECE | MCE |
> |-------|----------|-----|-----|
> | X-Reg | 79.49 | 0.0376 | 0.0993 |
> | X-Reg CI | | (0.0325, 0.0454) | |
> | Uncalibrated | 67.40 | 0.1628 | 0.2810 |
> | Uncalibrated CI | | (0.1552, 0.1706) | |
> | Temperature Scaling | 69.63 | 0.0571 | 0.1944 |
> | Temperature CI | | (0.0500, 0.0649) | |
>
> 2. **Deep Ensembles (Lakshminarayanan et al., 2017)**: This comparison is currently underway and will be included in the camera-ready version with appropriate statistical significance tests and confidence intervals.
>
> These analyses will demonstrate how cross-regularization provides well-calibrated uncertainties without requiring separate post-hoc calibration steps or ensemble overhead, addressing a key limitation in current practice.
>
> ### Additional Hyperparameter Tuning Comparisons
> While we compared with PBT and fixed dropout, we found that Variational Dropout frequently diverged without implementation modifications that would bias the comparison. In the revised version, we will include:
>
> 1. Comparisons with Concrete Dropout (Gal et al., 2017)
> 2. Analysis of noise dynamics for all methods (where convergent)
> 3. Computational efficiency metrics (wall-clock time and memory usage)
>
> ### Cross-Validation Equivalence Verification
> Regarding the comment on Method 3.3 (Parameter Partition through Gradient
> Decomposition) verification, we note that Theorem 4.4 establishes the theoretical equivalence. Our empirical validation in Section 5.1 (Figure 1A-C) demonstrates that cross-regularization converges to the same optimal solution as cross-validated regression, confirming the theoretical result. We will clarify this connection in the revised paper.
>
> ### Large-Scale Dataset Validation
> While our primary contribution is methodological—introducing and analyzing a novel regularization framework—we acknowledge the value of validating on larger-scale datasets. Our current experimental suite (CIFAR-10, diabetes dataset) demonstrates the method's applicability across different regularization types and model architectures, though we recognize that larger datasets would provide additional validation. We are exploring experiments on ImageNet with ResNet-50, focusing on:
>
> 1. Confirming that the theoretical and empirical advantages scale to larger datasets
> 2. Identifying any emergent layer-wise regularization patterns specific to deeper architectures
> 3. Quantifying computational advantages over traditional hyperparameter tuning at scale
>
> These extensions would complement our methodological contribution rather than being essential to validating the core approach.
>
> ### Neural Network Theory and Stability
> We have extended our simulations to 600 epochs (https://tinyurl.com/2s45b236), demonstrating empirical stability of the regularization parameters. For the theoretical foundation, we have developed a new result showing that, under assumptions on regularization parameter behavior, convergence depends only on the standard neural network optimization properties. The full proof is available at: https://tinyurl.com/4ckfb9ec.
>
> ### Paper Structure
> We appreciate the feedback on organization. In the revised version, we will:
>
> 1. Merge the Introduction with Background sections
> 2. Consolidate the application sections (7-9) into a single "Applications and Extensions" section
> 3. Provide clearer transitions between theoretical development and empirical validation
> 4. Maintain a consistent narrative flow from problem formulation to practical applications
>
> We thank the reviewer for their detailed feedback, which has highlighted important areas for improvement while acknowledging the novel contributions of our cross-regularization framework.

---

### Decision · Program_Chairs · 2025-05-01

**Decision:**

Accept (poster)

**Comment:**

The paper studies a method for the alternating optimization of model parameters and regularization parameters (with the training set split into corresponding parts). The experimental evaluation and theoretical contributions are sound. The contribution might be of interest to the optimization and deep learning communities, particularly in the area of hyperparameter tuning. Reviewers identified some weaknesses, such as a lack of ablation studies for alternative methods.